# Bio-inspired poly-DL-serine materials resist the foreign-body response

Donghui Zhang [1], Qi Chen [2], Yufang Bi[2], Haodong Zhang [2], Minzhang Chen[2], Jianglin Wan[2], Chao Shi[2], Wenjing Zhang [2], Junyu Zhang[2], Zhongqian Qiao[2], Jin Li[3], Shengfu Chen[4] & Runhui Liu [1,2✉]

Implantation-caused foreign-body response (FBR) is a commonly encountered issue and can result in failure of implants. The high L-serine content in low immunogenic silk sericin, and the high D-serine content as a neurotransmitter together inspire us to prepare poly-DL-serine (PSer) materials in mitigating the FBR. Here we report highly water soluble, biocompatible and easily accessible PSer hydrogels that cause negligible inflammatory response after subcutaneous implantation in mice for 1 week and 2 weeks. No obvious collagen capsulation is found surrounding the PSer hydrogels after 4 weeks, 3 months and 7 months post implantation. Histological analysis on inflammatory cytokines and RNA-seq assay both indicate that PSer hydrogels show low FBR, comparable to the Mock group. The anti-FBR performance of PSer hydrogels at all time points surpass the poly(ethyleneglycol) hydrogels that is widely utilized as bio-inert materials, implying the potent and wide application of PSer materials in implantable biomaterials and biomedical devices.

[1] State Key Laboratory of Bioreactor Engineering, East China University of Science and Technology, Shanghai, China. [2] Key Laboratory for Ultrafine Materials of Ministry of Education, Frontiers Science Center for Materiobiology and Dynamic Chemistry, Research Center for Biomedical Materials of Ministry of Education, School of Materials Science and Engineering, East China University of Science and Technology, Shanghai, China. [3] Shanghai Key Laboratory of Orbital Diseases and Ocular Oncology, Department of Ophthalmology, Ninth People's Hospital, Shanghai Jiao Tong University School of Medicine, Shanghai, China. [4] Key Laboratory of Biomass Chemical Engineering of Ministry of Education, College of Chemical and Biological Engineering, Zhejiang University, Hangzhou, Zhejiang, China. ✉email: rliu@ecust.edu.cn

Implantable biomaterials and devices will be recognized by the host immune system as foreign objects to initiate a series of interactions at the implant-host interface, which leads to the foreign-body response (FBR) including strong inflammatory responses, foreign-body giant cell formation, fibrosis, and eventually collagen encapsulation around the implants and isolation of implants from the host tissue[1–12]. The FBR of implants will lead to cutting off of nutrient transportation, painful tissue deformation, and even implant failure. Divergent from the need of anti-FBR requirements in clinical practice, a very limited number of anti-FBR materials were reported, and developing low FBR materials is still a formidable challenge[13–26]. Moreover, the increasing clinical demand on implantable biomaterials and medical devices urgently calls for high-efficiency anti-FBR materials with easy and scalable synthesis, low cost, and high solubility in water to avoid using organic solvent during material processing for biomedical application. For example, artificial vitreous needs to be molded in situ in the eye[23], and pancreatic islets encapsulation devices for diabetes therapy often need to be prepared in situ in a hydrogel precursor solution[27]. The preparation process of these hydrogel precursors must have a high degree of water solubility, and the high water solubility is also a prerequisite for anti-FBR materials in drug delivery and protein conjugation[28–31].

Currently, poly(ethyleneglycol) (PEG) has been extensively used as a biologically inert material[32–35], however, recent studies have revealed both immunogenicity and antigenicity of PEG and PEG-modified objects[36–38]. Another shortcoming of PEG is that it decomposes in the presence of oxygen in physiological conditions[39]. These shortcomings of PEG have limited its long-term in vivo application for biomaterials and biomedical devices. It is imperative to develop anti-FBR materials because materials have been the bottleneck that greatly impedes the advance in this field and the development of implantable biomaterials and medical devices[11,13]. Inspired by the low immunogenic silk sericin, we speculated that poly-L-serine (P-L-Ser) materials may resist the FBR. However, P-L-Ser is not soluble (<0.1 mg/mL) in water due to its folding into β-sheet[40,41], which greatly limits the application of P-L-Ser (Fig. 1a, b). To address the solubility issue, we recently explored poly-β-homoserine (β-HS) as a proof-of-concept demonstration of antifouling and anti-FBR materials with "dual hydrogen bonding hydration" (Fig. 1a, b)[38]. Although β-HS has outstanding anti-FBR performance, β-HS hydrogels were prepared using dimethylsulfoxide as solvent due to its moderate water solubility (about 10 mg/mL) (Fig. 1b)[38], which limit its application in implantable biomaterials such as artificial vitreous and cell encapsulation therapy.

To address the aforementioned challenge, we turn our attention to poly-DL-serine (PSer) that has a structure similar to P-L-Ser (α-DL- vs. α-L-amino acid) and β-HS (α- vs. β-amino acid), and also follows the "dual hydrogen bonding hydration" principle in designing anti-FBR materials. We introduced D-serine to prepare anti-FBR PSer because D-serine is one of the highest-level D-amino acids in the human body as a neurotransmitter[42–44], suggesting PSer (having α-L-amino and α-D-amino at 1:1 ratio) as a type of highly biocompatible and low immunogenic anti-FBR material (Fig. 1a). In addition, PSer has a high solubility in water (>500 mg/mL) because the incorporation of D-serine breaks the tendency of folding into β-sheet and affords PSer a random coil structure (Fig. 1a–c). Moreover, the synthesis of β-HS is operated by multi-step synthesis and under strictly anhydrous conditions for polymerization in a glove box using dry solvent, which indicates that β-HS is not ideal for scale-up synthesis and cost-effective application (Fig. 1d). In sharp contrast, both PSer monomer and PSer polymer are obtained from a simple one-step reaction in high yield, insensitive to water,

and operated in open vessels under ambient condition[40,45], which indicates the easy scale-up synthesis and promising application of PSer (Fig. 1d).

In this study, we synthesized poly-DL-serine diacrylamide (PSerDA) that enables the facile preparation of PSer hydrogels via photo-crosslinking (Fig. 1e). Subcutaneous implantation in mice demonstrated that PSer hydrogels display superior anti-FBR properties than that of PEG hydrogels by showing low inflammatory response and low expression of related inflammatory cytokines and genes after implantation within the first 2 weeks, and no obvious collagen wrapping after implantation for at least 7 months (Fig. 1f).

## Results

**Preparation of PSer hydrogels**. We synthesized two poly-DL-serine diacrylamide (PSerDA, $M_n = 3300$ Da, DP = 36; and 8600 Da, DP = 97) from O-tert-butyl-DL-serine over three steps, as hydrogel precursors with narrow dispersity (Đ = 1.12–1.18, Fig. 2a–c and Supplementary Table 1) to prepare two different PSer hydrogels, PSer3300, and PSer8600, and implanted them into mice. Two poly(ethyleneglycol) diacrylate (PEGDA) were used to prepare PEG hydrogels, PEG2000 and PEG5000, for comparisons. Hydrogels of PSer and PEG were prepared by UV-initiated gelation of 20 wt% PSerDA and PEGDA solutions, respectively, using Irgacure 2959 as the photoinitiator (Supplementary Table 2). The Fourier transform infrared (FTIR) spectra of hydrogels and their precursors indicated complete gelation based on the disappearance of characteristic peaks for PEGDA at 1410, 1195, and 810 cm$^{-1}$ in the PEG hydrogels, and the disappearance of characteristic peaks for PSerDA peaks at 1402, 1203, and 802 cm$^{-1}$ in the PSer hydrogels (Fig. 2d and Supplementary Fig. 1). We found that X-ray photoelectron spectroscopy (XPS) spectra indicated functional groups of PSer hydrogels, including C–C, C–N, and O = C–N peaks, and the C–C peak ratio of PSer3300 hydrogel is higher than that of PSer8600 hydrogel, indicating that the former has a higher degree of crosslinking (Fig. 2e, Supplementary Fig. 2 and Supplementary Table 3). Scanning electron microscopy (SEM) characterization showed that all PEG and PSer hydrogels have porous structures (Fig. 2f).

**PSer hydrogels mitigate the inflammatory response and the FBR**. We evaluated the tissues surrounding PEG and PSer hydrogels after implantation for 1 week and 2 weeks and found that the tissue surrounding PEG hydrogels, rather than PSer hydrogels, is swollen and inflammatory, which is more severe after 1 week than after 2 weeks (Fig. 3a, b). Histological analysis using hematoxylin-eosin (H&E) staining showed that PEG2000 hydrogels and PEG5000 hydrogels caused an obvious inflammatory reaction in the surrounding tissues at 1 week, and a slightly reduced, but still obvious, inflammatory reaction at 2 weeks. F4/80 immunofluorescent staining also revealed a mass of macrophages in tissues surrounding PEG2000 and PEG5000 hydrogels at both 1 week and 2 weeks. The surrounding tissues of PSer3300 hydrogels and PSer8600 hydrogels both showed substantially lower inflammatory reaction and lower density of macrophages than that of PEG hydrogels at both 1 week and 2 weeks post-implantation, which indicated that the PSer hydrogels are superior to PEG hydrogels, regarding compatibility to the host.

We continued to evaluate these hydrogel implants after 4 weeks of implantation. Masson's trichrome (M&T) staining showed that both PEG hydrogels were surrounded by a dense layer of inflammatory cells, around 27 μm for PEG2000 and around 22 μm for PEG5000, which is comparable to previous reports[46,47];

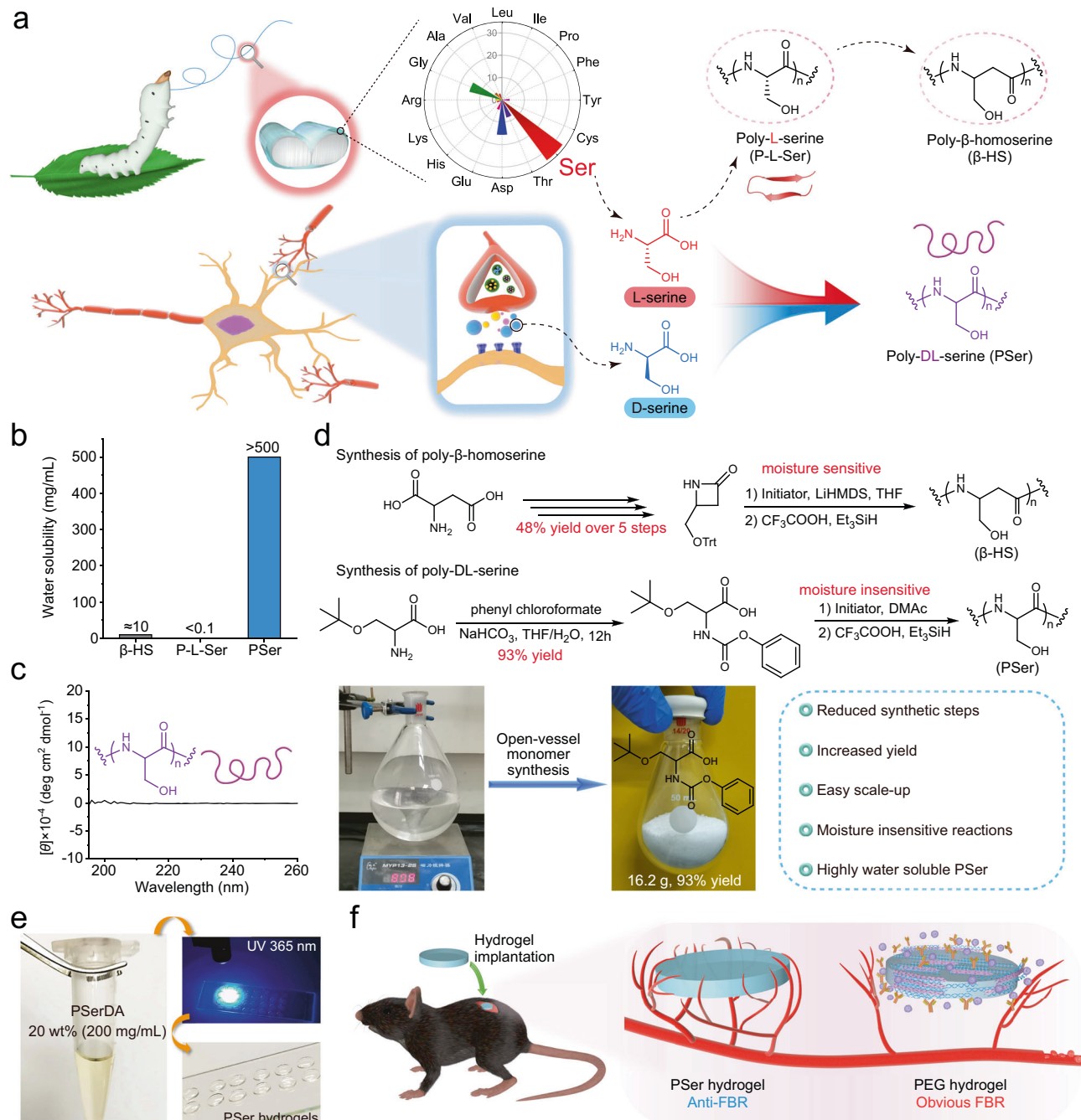

**Fig. 1 Design of PSer hydrogels with low FBR. a** Design of poly-DL-serine (PSer) from L-serine and D-serine. The high L-serine content in silk sericin and the high level of D-serine in the human body as an important neurotransmitter altogether inspired the design of anti-FBR material PSer. **b** Water solubility of poly-β-homoserine (β-HS) (about 10 mg/mL), poly-L-serine (P-L-Ser) (<0.1 mg/mL due to its β-sheet folding) and PSer (>500 mg/mL). **c** Circular dichroism spectrum of PSer. **d** Synthesis of β-HS and PSer. LiHMDS Lithium hexamethyldisilazide, DMAc dimethylacetamide. **e** Photographs of poly-DL-serine diacrylamide (PSerDA) that was well dissolved at a concentration of 20 wt% and was used to prepare PSer hydrogels by photo-crosslinking in the presence of 0.1% photoinitiator (Irgacure 2959). **f** PSer hydrogels and PEG hydrogels implanted subcutaneously into C57/BL6 mice induced low FBR and obvious FBR respectively.

in sharp contrast, both PSer hydrogels were surrounded by tissues with only a thin layer of inflammatory cells (< 5 μm thickness) at the hydrogel-host tissue interface (Fig. 4a, b). The collagen density at the interface of the two PSer hydrogels and tissues (62–76%) was significantly lower than that of both PEG hydrogels (>90%) (Fig. 4c). Both M&T and α-smooth muscle actin (αSMA) staining indicated a higher density of blood vessel in the tissue surrounding the PSer hydrogels (126 and 106 blood vessels per mm$^2$ for PSer3300 hydrogels and PSer8600 hydrogels, respectively) than

that in the tissue surrounding the PEG hydrogels (26 and 33 blood vessels per mm$^2$ for PEG2000 hydrogels and PEG5000 hydrogel, respectively) (Fig. 4d). The higher density of blood-vessels around PSer hydrogels than that around the PEG hydrogels indicated highly favored substance exchanges, such as nutrients and oxygen transportation around the PSer hydrogels implants.

After implantation for 3 months, when the stage of inflammatory response passed, we found both PEG hydrogels

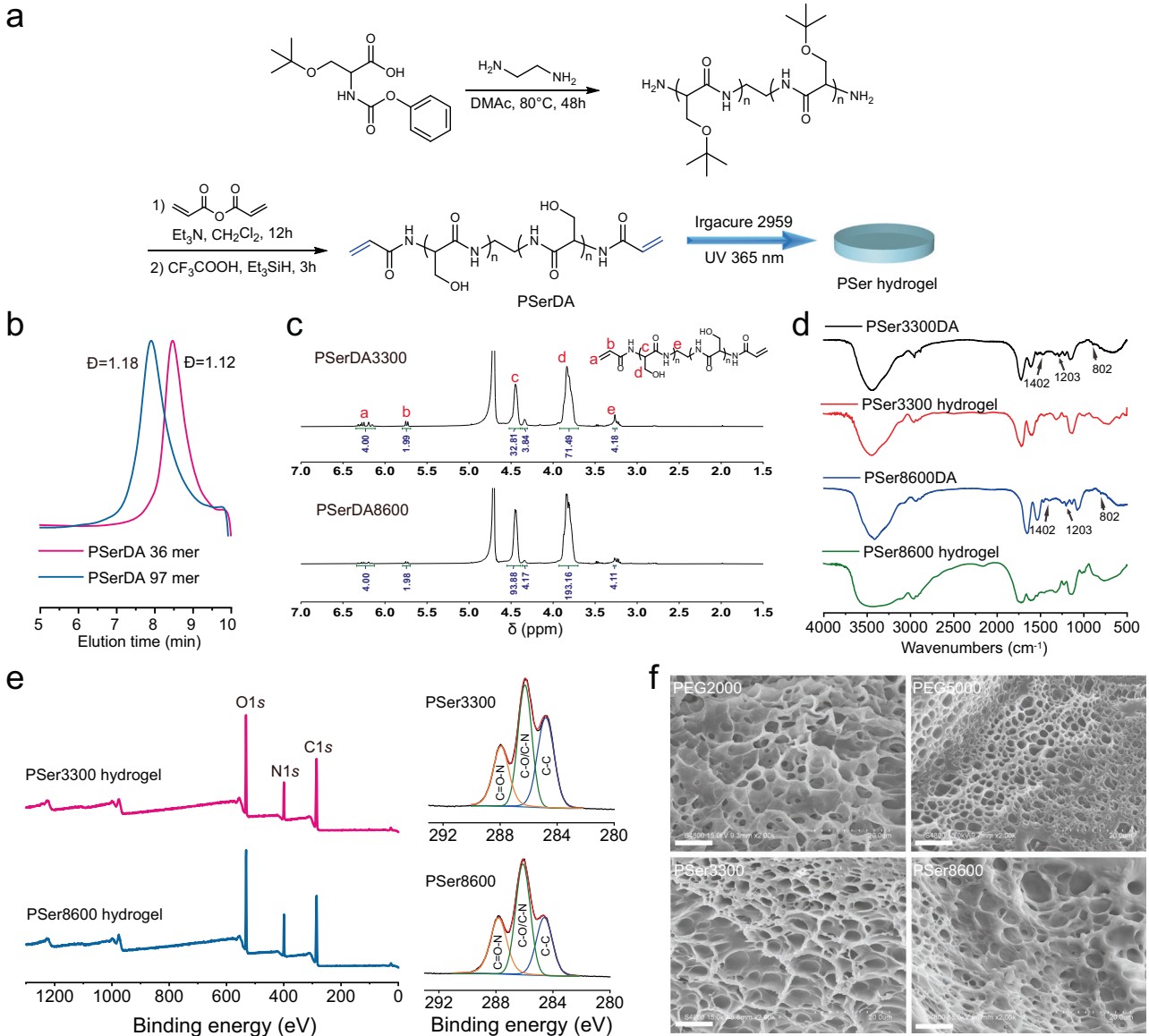

**Fig. 2 Preparation and characterization of PSer hydrogels. a** Synthesis of PSer hydrogel precursor PSerDA and preparation of PSer hydrogels. **b** GPC traces of two PSerDA at a different length, using THF as the mobile phase at a flow rate of 1 mL/min. **c** $^1$H NMR spectra (400 MHz, D$_2$O) of PSerDA. **d** FTIR spectra of PSerDA and PSer hydrogels. **e** XPS spectra of PSer hydrogels. **f** SEM images of hydrogels, scale bar: 20 μm.

were still encapsulated by a dense layer of inflammatory cells at around 19 μm thickness and a dense capsule layer with >90% collagen density, while no obvious layer of inflammatory cells was found around the two PSer hydrogels, with diffuse collagens that the density was decreased to 48–69% (Fig. 4e–g). Meanwhile, the blood vessel density surrounding the PSer hydrogels was still much higher than that surrounding the PEG hydrogels, a trend similar to that after 4 weeks implantation aforementioned (Fig. 4h). The above analysis after implantation for 1 week to 3 months indicated that PSer hydrogels with different cross-linking densities and precursor molecular weight, PSer3300, and PSer8600, have similar and favorable results to mitigate the FBR, which indicated that the anti-FBR performance of the PSer hydrogels likely depends on the material property itself. In a continuous study for 7 months of implantation, we focused on PSer3300 hydrogels using PEG2000 hydrogels for comparison. We found that the PSer hydrogels were only surrounded by sparse collagen, however, the PEG hydrogels were surrounded by dense fibrous capsules (Fig. 4i–j). The long-term implantation

study indicated that PSer is a class of promising anti-FBR material.

We also measured the stiffness of PEG and PSer hydrogels by compressive test and found that PEG hydrogels have a stronger compression modulus (1.07 MPa for PEG2000 and 0.582 MPa for PEG5000) than PSer hydrogels (0.121 MPa for PSer3300 and 0.046 MPa for PSer8600) (Supplementary Fig. 3), though we prepared all these hydrogels using 20 wt% PSerDA and PEGDA solutions. These hydrogels made from 20 wt% precursors have similar water content, such as the comparable water content between PEG2000 (~84.0%) and PSer3300 (~83.0%), or between PEG5000 (~88.5%) and PSer8600 (~86.4%) (Supplementary Fig. 4). The difference in mechanical properties between PEG and PSer hydrogels may be attributed to the super-hydrophilic structure of PSer because the hydrophilic characteristics will lead to weak mechanical properties of hydrogels[48]. Since mechanical properties could have an impact on the outcome of the anti-FBR function of materials[11], we prepared an even softer PEG hydrogel

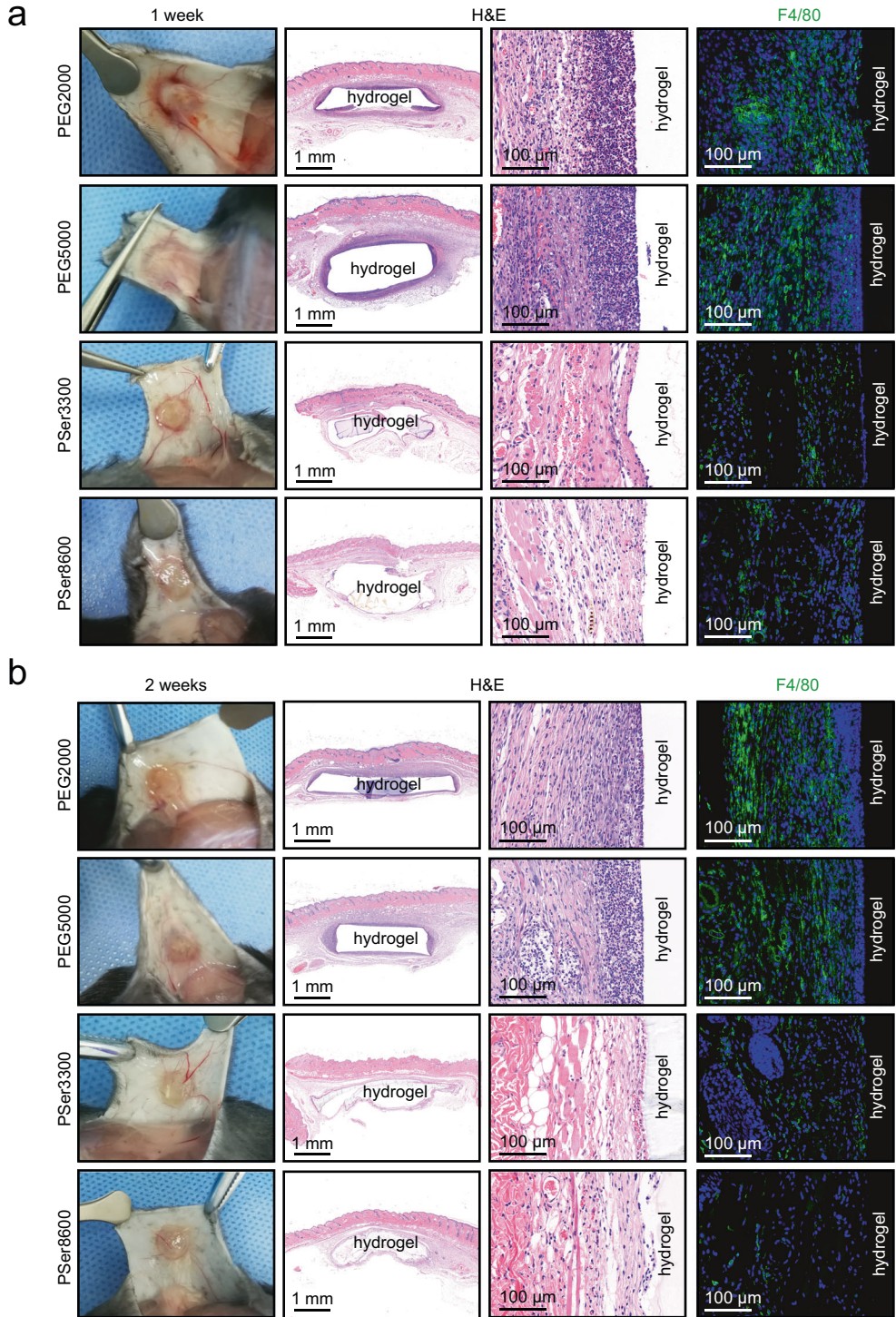

**Fig. 3 Inflammatory response in tissues surrounding the PEG and PSer hydrogels.** Explantation and staining images of PEG and PSer hydrogels after subcutaneously implanted in mice for 1 week (**a**) and 2 weeks (**b**). PEG hydrogels visually were observed to be blurry and full of cell and tissue deposition, while PSer hydrogels were observed to be more transparent and devoid of cellular deposition. The inflammatory response to the implants were evaluated by a hematoxylin-eosin (H&E) stain. Macrophages were labeled by pan macrophage immunofluorescent biomarker (F4/80) and were stained in green fluorescence. Nucleus were stained by DAPI to show blue fluorescence.

(PEG5000-H hydrogel) for implantation study, using only 4 wt % of PEG5000DA as a precursor. The PEG5000-H hydrogel has a high water content (~96.0%) and a low compressive modulus (0.017 MPa), which is much weaker than both PSer hydrogels (Supplementary Fig. 3 and Supplementary Fig. 4).

After 1 week of implantation of PEG5000-H hydrogels, the hydrogels were wrapped with a layer of inflammatory cells, which is weaker than PEG2000 and PEG5000 in causing the body's inflammatory response. Although the mechanical properties of PEG5000-H hydrogels are only 1/7.2 of PSer3300 and 1/2.8 of

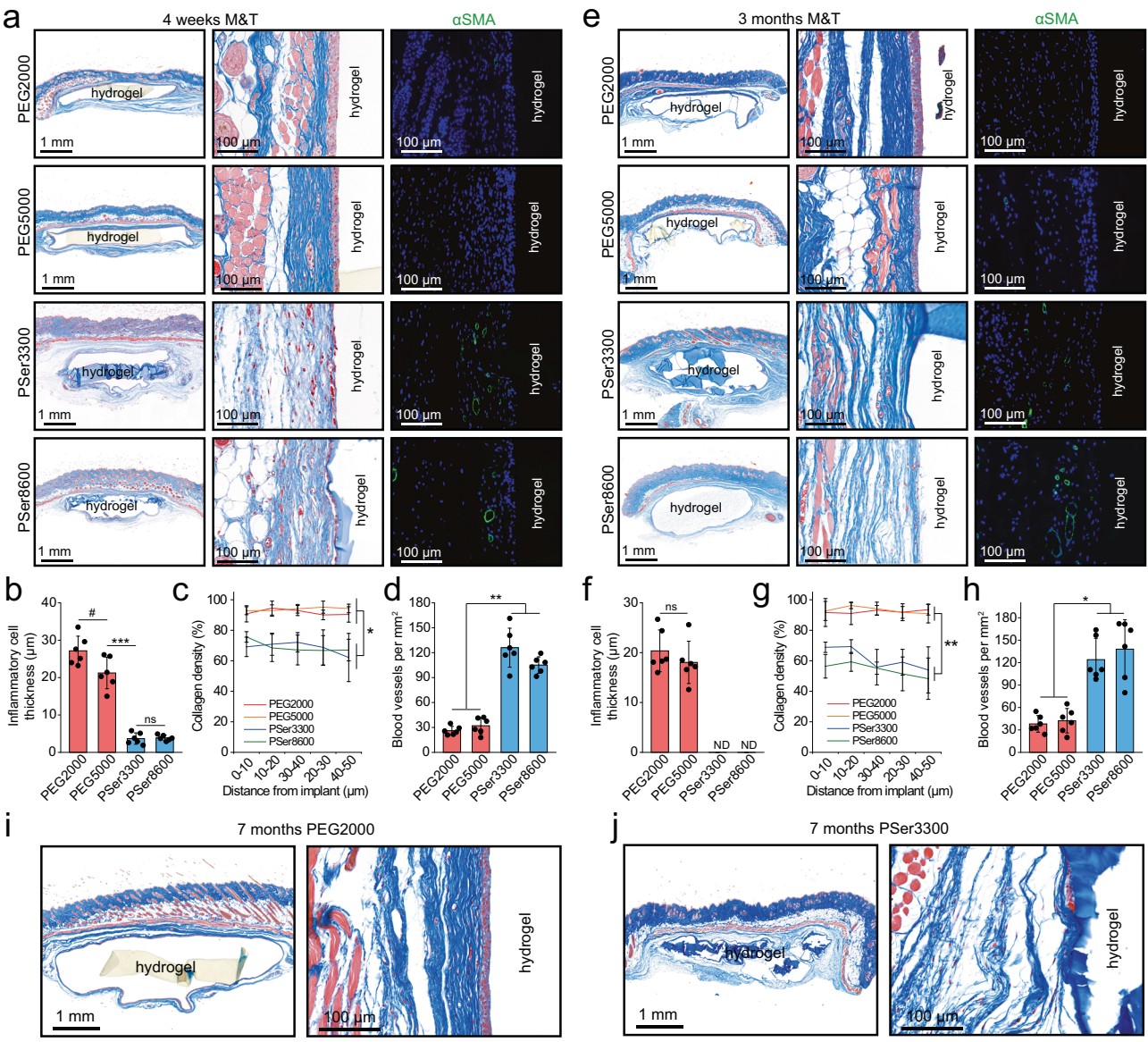

**Fig. 4 Collagen and blood vessel density in tissues surrounding the PEG and PSer hydrogels 4 weeks and 3 months post subcutaneous implantation. a** Masson's trichrome (M&T) staining and α-smooth muscle actin (αSMA) immunofluorescent staining to evaluate collagen encapsulation and blood vessels (green fluorescence) density after 4 weeks of implantation. Nucleus were stained by DAPI to show blue fluorescence. **b–d** Quantified data of inflammatory cell thickness (**b**), collagen density (**c**), and blood vessel density (**d**) ($n = 6$, mean values ± sd) of the hydrogel-tissue interface after 4 weeks of implantation. **e** M&T staining and αSMA immunofluorescent staining to evaluate collagen encapsulation and blood vessels (green fluorescence) density after 3 months of implantation. Nucleus were stained by DAPI to show blue fluorescence. **f–h** Quantified inflammatory cell thickness (**f**), collagen density (**g**), and blood vessel density (**h**) ($n = 6$, mean values ± s.d.) of the hydrogel-tissue interface after 3 months of implantation. ND represents not detected. **i, j** M&T staining of tissues surrounding PEG (**i**) and PSer (**j**) hydrogels after implantation for 7 months. Statistical analysis: one-way ANOVA with Tukey post-test, [#]$p < 0.05$, [*]$p < 0.01$, [**]$p < 0.001$, [***]$p < 0.0001$, ns: not significant.

PSer8600, the inflammatory response caused by PEG5000-H hydrogels are still stronger than that of both PSer hydrogels (Supplementary Fig. 5a). After 4 weeks of implantation, PEG5000-H hydrogels were encapsulated by a dense layer of inflammatory cells with fibrosis, resulted in obvious FBR (Supplementary Fig. 5b). Quantification on the collagen density at the interface of the PEG5000-H hydrogels and tissues gave a ratio of 87–91% that is similar to the collagen density around PEG2000 and PEG5000 hydrogels and is much higher than the collagen density around PSer3300 and PSer8600 hydrogels (Supplementary Fig. 5c). Therefore, the above studies on the mechanics of the implant, using the stiffness, support our

conclusion and the merit of our PSer hydrogels as promising anti-FBR materials.

Although PEG hydrogels are typically characterized as bioinert, PEG-based materials can promote a degree of complement activation in vivo[49], so we also used polyacrylamide (PAM) hydrogels as control, as it has similar water content, but is not particularly a complement activator. We synthesized two polyacrylamide hydrogels (PAM1 and PAM2) using 20 wt% precursors with different crosslinking density (Supplementary Table 2). After subcutaneous implantation in mice, both PAM hydrogels (PAM1 and PAM2) caused obvious inflammatory response after 1 week and collagen capsulation after 4 weeks, as

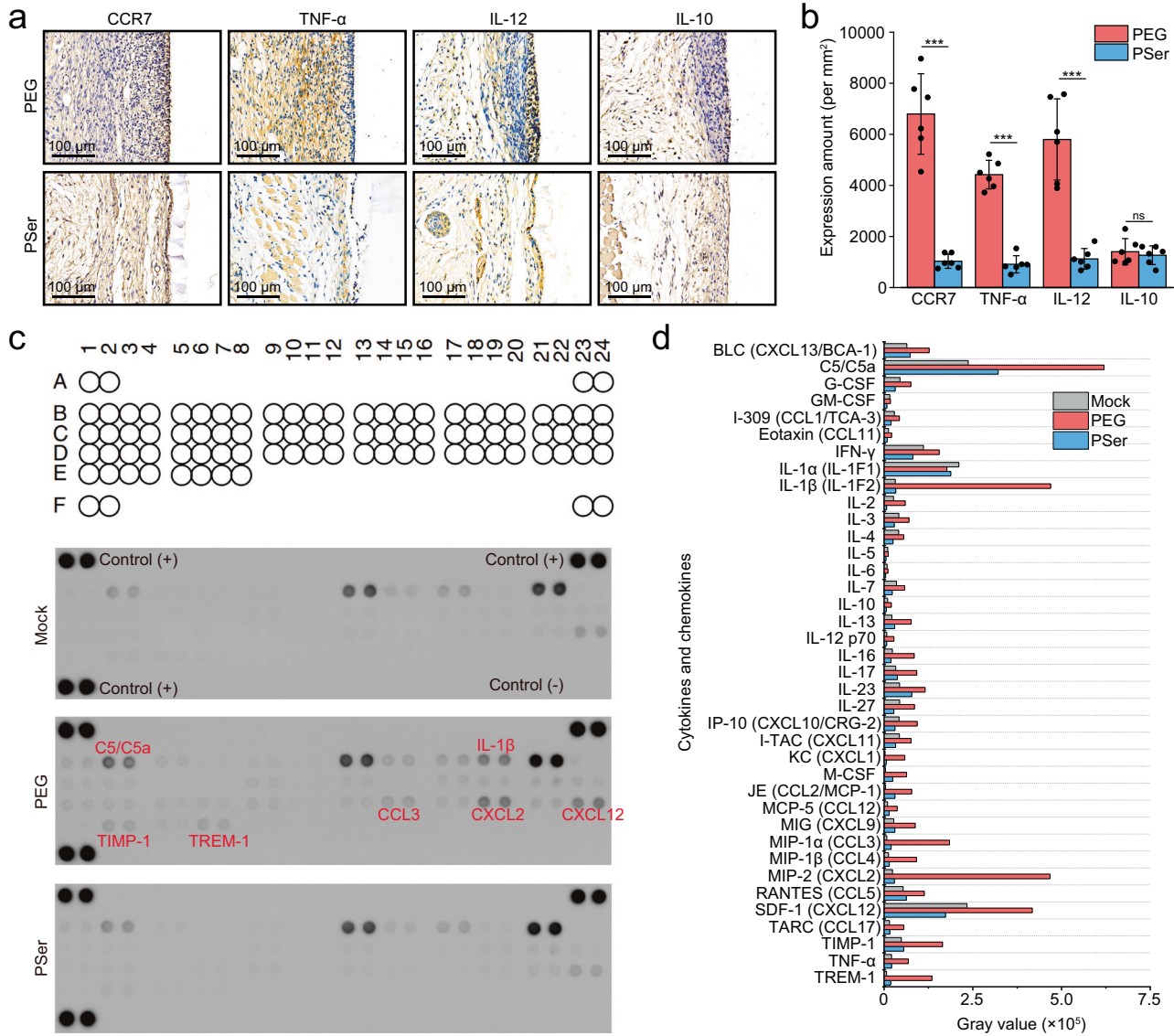

**Fig. 5 Inflammatory marker staining and cytokine profiling analysis 2 weeks post implantation. a** Immunohistochemical staining of inflammatory markers (CCR7, TNF-α, IL-12, and IL-10) in tissues surrounding PEG2000 and PSer3300 hydrogels. Cells stained by inflammatory markers show brown color, while all nucleus stained with hematoxylin show blue color. **b** Quantification of inflammatory marker expression. Data were collected in the tissue within 100 μm from the tissue-hydrogel interface. $n = 6$ (six mice for each type of hydrogel), mean values ± s.d. **c** Cytokine profiling for tissue lysates 2 weeks post-implantation of PEG2000 and PSer3300 hydrogels, using Mock group without implantation as control. The mouse cytokine array coordinates are shown in Supplementary Table 4. **d** Gray values of cytokine and chemokine expression. Statistical analysis: two-tailed $t$-test, $**p < 0.001$, ns: not significant.

shown in the histological staining (H&E for 1 week and M&T for 4 weeks, Supplementary Fig. 6). Therefore, despite the poly-acrylamide hydrogel is not a complement activator it caused obvious FBR after implantation. Our observation is consistent to other researchers' reports on the foreign-body granuloma of the glabella due to polyacrylamide filler[50].

**Inflammatory factor analysis.** To find out the reason for the low inflammatory response and FBR associated with PSer hydrogel implants, we analyzed the expression of inflammatory factors at 2 weeks after hydrogel implantation because at this time point the expression of different types of inflammatory factors promoted or regulated the FBR[7]. We stained the tissues surrounding the hydrogels for pro-inflammatory markers CCR7, TNF-α, and IL-12, and anti-inflammatory marker IL-10 to study the expression

of inflammatory factors 2 weeks after hydrogel implantation. The results showed that tissues surrounding the PSer hydrogels have low expression of pro-inflammatory markers (CCR7, TNF-α, and IL-12) that is only about 1/4 to 1/6 fold of these marks in tissues surrounding the PEG hydrogels; whereas, the expression of anti-inflammatory marker (IL-10) in tissues surrounding the implants are at low level and without significant difference between the PSer hydrogels and the PEG hydrogels (Fig. 5a, b). This result indicates that the anti-FBR property of PSer hydrogels has a strong correlation to the low expression of pro-inflammatory factors that plays a significant role in the process of the FBR and triggers the collagen encapsulation to the implants[6].

Inspired by the result of the above inflammatory factor analysis, we continued to explore the expression of 40 different inflammation-related cytokines and chemokines in tissues surrounding the hydrogel implants 2 weeks post-implantation,

using proteome profiler antibody arrays. The tissue sample surrounding the PSer hydrogels at 2 weeks post-implantation had an expression of inflammatory factors comparable to that of the Mock group without implantation, and both had less inflammatory cytokines and chemokines than that surrounding the PEG hydrogels including C5/C5a, IL-1β, IP-10, CCL3, CCL4, TNF-α and TREM-1 (Fig. 5c, d). In addition, samples from PSer hydrogel implantation, compared to that from PEG hydrogel implantation, had substantially lower expression of the TIMP-1 that inhibits matrix metalloproteinase and promotes fibrosis, and the strong neutrophil chemotactic agents CXCL1, CXCL2, and CXCL12 that are involved in many immune responses[18]. The result echoed above H&E staining after 2 weeks post-implantation that tissues surrounding the PEG hydrogel still had an obvious inflammatory response, however, tissues surrounding the PSer hydrogel had a low inflammatory response and few macrophages.

**Gene expression analysis.** We continued to analyze the gene expression by RNA-seq in tissues surrounding the hydrogels 2 weeks post-implantation, using a Mock group without implantation as the control. Under the condition of |log_2 fold changes|>1 and Q value <0.001, the differential gene number between samples of PEG hydrogels and Mock group was 2637; while the differential gene number between samples of PSer hydrogel and Mock group was 1603. Analysis of these two differential gene groups based on Kyoto Encyclopedia of Genes and Genomes (KEGG) pathway enrichment revealed a higher number of differential genes, smaller Q value, and greater rich ratio of the PEG/Mock comparison than the PSer/Mock comparison, regarding inflammatory reaction-associated pathways such as cytokine-cytokine receptor interaction, NOD-like receptor signaling pathway, and chemokine signaling pathway (Fig. 6a, b). We also did the KEGG pathway classification analysis and found that the PEG/Mock comparison generally had a higher number of differential genes than the PSer/Mock comparison, within which we observed 304 and 160 differential genes respectively for the PEG/Mock comparison and PSer/Mock comparison in the immune system (Fig. 6c).

We then did further analysis on the immune system using the gene ontology (GO) classification analysis and found a significant difference between the two groups, PEG/Mock comparison, and PSer/Mock comparison. Under conditions of |log_2 fold changes|>1 and Q value <0.001, we found 536 and 276 upregulated genes respectively for the PEG hydrogel and PSer hydrogel implantation samples (Fig. 6d, e), and among the top 30 upregulated genes listed in the table we found an overall higher expression of the upregulation genes induced by PEG hydrogels than did the PSer hydrogels (Fig. 6f). We analyzed the expression level of 109 immune-related genes, including genes associated with M1 macrophages, M2 macrophages, cytokines, and T cells, and found from the heat map that PEG hydrogels implantation induced more prominent expression on M1 type macrophage, cytokine, and T cell-associated genes than did the PSer hydrogels implantation (Fig. 6g). For the M2-type macrophage-related genes that are known to regulate the FBR of implantation, we observed similar expression in most genes between the PEG hydrogel and PSer hydrogel implantation samples. It's noteworthy that among these 109 immune-related genes, the change of most of the gene expression induced by PSer hydrogel implantation was negligible compared to the Mock group without implantation.

To further evaluate the innate immune response to PEG and PSer hydrogel implantation, we did a qPCR analysis on the tissue surrounding the hydrogels 2 weeks post-implantation

(Supplementary Fig. 7). The result showed that PEG hydrogel implantation induced upregulated gene expression in *Col1a1* (collagen marker), *CD11b* (myeloid cell marker), *CD68* (macrophage marker,) *Ly6g* (neutrophil marker), and *Ccl19* (M1 macrophage marker), 3.0–7.8 folds higher than did the PSer hydrogel implantation. In sharp contrast, PSer hydrogel implantation induced upregulated gene expression in *Cxcl13* (M2 macrophage marker) at 3.8 folds higher than did the PEG hydrogel implantation. These results underpinned the conclusion in the RNA-seq that PSer hydrogels induces lower pro-inflammatory gene expression than does the PEG hydrogel, and could cause upregulation of some immunoregulatory genes.

In summary, we design poly-DL-serine (PSer) as a class of anti-FBR materials that are composed of L-serine and D-serine, which is inspired by the high L-serine content in silk sericin and the high level of D-serine in the human body as an important neurotransmitter altogether. We prepared anti-FBR hydrogels from PSer that is obtained through a simple synthesis and is highly soluble in water. PSer hydrogel implants in mice shows low inflammatory response after 1 week and 2 weeks, and resist collagen capsules for at least 7 months. The inflammatory factor and gene expression of tissues surrounding PSer hydrogels are comparable to the Mock group without implantation, whereas, high expression of cytokine and upregulation of pro-inflammatory gene are obvious in tissues surrounding PEG hydrogels, which implies the low FBR property of PSer hydrogels. The promising anti-FBR property and easy accessibility of the PSer hydrogels suggest their wide application in implantable biomaterials and biomedical devices.

## Methods

**Measurements.** The molecular weight of the compound was performed on a Waters XEVO G2 TOF mass spectrometer with MassLynxTM software. Gel permeation chromatography (GPC) was performed on a Waters GPC instrument (with Breeze 2 software) equipped with a Waters 1515 isocratic HPLC pump and a refractive index detector (Waters 2414), using tetrahydrofuran (THF) as the mobile phase at a flow rate of 1 mL/min at 40 °C. The GPC were equipped by Styragel HR 3 (particle size 7.8 µm) columns linked in series. Relative number-average molecular weight (Mn) and dispersity index (Đ) were calculated from a calibration curve using polystyrene (PS) as standards. $^1$H NMR spectra were collected on an AVANCE III 400 spectrometer with TopSpin software at 400 MHz, using CDCl$_3$ or D$_2$O as the solvent. $^1$H NMR chemical shifts were referenced to the resonance for residual protonated solvent (δ 0.00 for TMS in CDCl$_3$ and 4.79 for D$_2$O). Circular dichroism (CD) spectroscopy was carried out on an Applied Photophysics Chirascan CD spectrometer with Pro-Data Viewer software. A solution of 0.2 mg/mL poly-DL-serine in H$_2$O was placed in a quartz cell with a light path of 1.0 mm. Fourier transform infrared (FTIR) spectra were recorded on a Thermo Electron Nicolet 6700 FTIR spectrophotometer by using a KBr plate. X-ray photoelectron spectroscopy (XPS) was conducted on a Thermo Scientific$^{TM}$ K-Alpha$^{TM}$ spectrometer equipped with a monochromatic Al Kα X-ray source (1486.6 eV) operating at 100 W. All peaks were calibrated with C1s peak binding energy at 284.8 eV for adventitious carbon. The experimental peaks were fitted with Avantage software. Scanning Electron Microscope (SEM) images were collected on a Hitachi Limited S-4800 Field Emission SEM with FE-PC SEM software. Information about other instruments and equipment is in the specific experimental part.

**Hydrogel preparation and characterization.** Hydrogels were prepared using a method similar to the previously reported procedure[38]. In brief, a solution of hydrogel precursor (Supplementary Table 2.) was sonicated till the solution turned to transparent. An aliquot of 12 µL of this solution was pipetted into each well (4.5 mm in diameter) on a chambered polydimethylsiloxane coverslip. Gelation proceeded under UV irradiation (365 nm) at a power of 500 mW/cm$^2$ for 3 min. Then, the hydrogels were removed from the mold and soaked in Millipore water at room temperature with exchange of freshwater every 12 h (total 5 times water change) to remove unreacted precursors and to reach an equilibrium of swelling state.

The lyophilized hydrogels were used for FTIR and XPS analysis. The swollen hydrogels were cooled and fractured in liquid nitrogen, and lyophilized for SEM assay. The water content of hydrogels was measured and calculated based on their mass before and after lyophilization ($n = 3$) using the equation below. Water content (%) = $(m_w − m_d)/m_w × 100$, where $m_w$ and $m_d$ are the mass of the wet hydrogel and the dry hydrogel, respectively. Hydrogels were prepared into 5-mm-diameter and 2-mm-thick cylinders ($n = 4$) for compressive test[48]. The

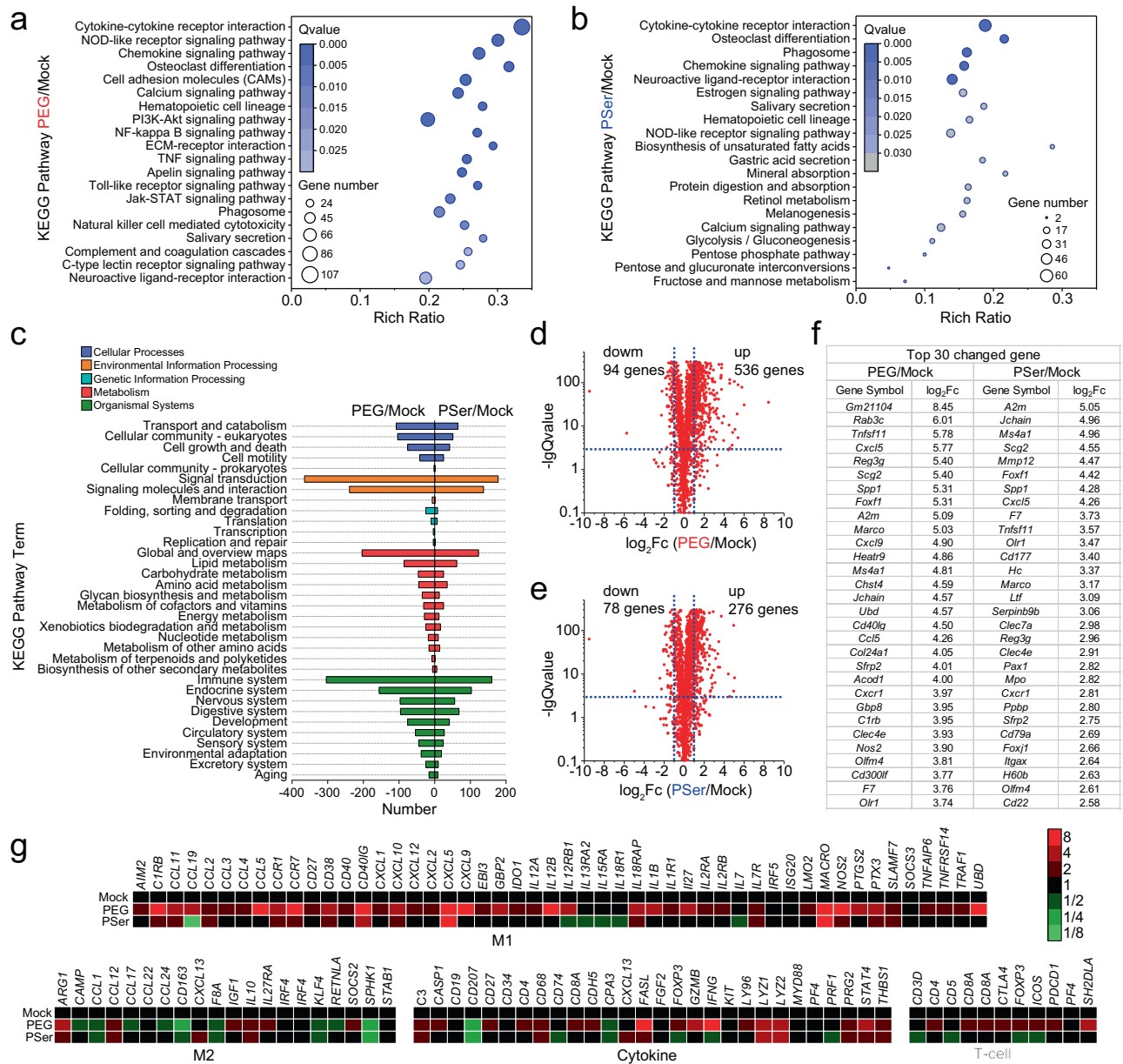

**Fig. 6 RNA-seq 2 weeks post implantation. a, b** KEGG pathway enrichment analysis of PEG/Mock comparison (**a**) and PSer/Mock comparison (**b**). **c** KEGG pathway classification analysis of PEG/Mock comparison and PSer/Mock comparison. **d, e, f** Significant gene expression changes of PEG/Mock comparison (**d**) and PSer/Mock comparison (**e**) in genes involved in the immune system as measured by GO classification analysis, and a list of top 30 upregulated genes and corresponding log₂Fc (log₂ fold change) (**f**). **g** Gene expression analysis of inflammatory factors in tissues surrounding the PEG and PSer hydrogels, with data normalized to the mock group (without implantation). $n = 3$ mice per treatment.

compressive property of hydrogels was tested on a universal testing machine (AI-3000, Gotech Testing Machines Co., Ltd., with U62 software) with a 100 N load cell at room temperature. The crosshead speed was set at 1 mm/min, and the test limit of compressive strain was set at 90% to protect the machine.

**Hydrogel disk implantation into mice and explantation**. All animal procedures were performed in accordance with the Guidelines for Care and Use of Laboratory Animals of the Ninth People's Hospital, and experiments were approved by the Animal Ethics Committee of the Ninth People's Hospital, Shanghai Jiao Tong University School of Medicine. Mice were raised in an IVC system at 20–26 °C and 40–70% humidity, with a dark/light cycle of 12 h. All hydrogels were tested for endotoxin using a Limulus Amebocyte Lysate (LAL) endotoxin assay kit (Solarbio, catalog no. T7574) prior to implantation. The hydrogel had negligible endotoxin, with a value far below the ISO 10993 standard -20 EU per implant material.

Hydrogel samples were swollen with sterile saline, sterilized by UV and implanted subcutaneously in C57BL/6 male mice at 6-week-old. Six replicates for each type of hydrogel were implanted into mice. Mice were anesthetized using pentobarbital and shaved. About an 8 mm longitudinal incision was made on the dorsal surface, using surgical scissors to provide access to the subcutaneous space. Then subcutaneous pockets on either side of the incision were created with blunt forceps for the implantation of the hydrogel disks. After implantation, the incisions were closed using wound clips. Mice were monitored until recovery from anesthesia and housed for 1 week, 2 weeks, 4 weeks, 3 months, or 7 months. The mice grew normally with no sign of discomfort after the implantation and no body weight loss was observed before explantation.

After each time point, mice were sacrificed and the hydrogel samples together with the surrounding tissue were excised and collected. The explanted samples were fixed in 4% paraformaldehyde overnight and embedded in paraffin wax. Sections of each sample at 3–5 μm thickness were cut and mounted onto slides for histological staining and imaging. All images were scanned in Pannoramic 250/MIDI-equipped with the CaseViewer 2.0 software.

All data are presented as a mean of biological replicates (six mice per type of hydrogel).

**Histological analysis**. The inflammatory response was examined by staining the tissue sections of 1 week and 2 weeks post-implantation with hematoxylin & eosin (H&E), which stains nuclei in blue and cell cytoplasm in pink. Collagen formation and distribution were examined by staining the tissue sections of 4 weeks, 3 months, and 7 months post-implantation with Masson's trichrome (M&T) that stains collagen in blue, cytoplasm in red, and nuclei in black. The inflammatory cell thickness is measured according to the thickness of the markedly red to purple layer at the hydrogel-tissue interface in the M&T images[51] ($n = 6$, mean values ± s.d.). The collagen density is measured by the percentage of blue-pixel coverage in the M&T images of tissues within 50 µm (at 10 µm steps) from the hydrogel-tissue interface ($n = 6$, mean values ± sd).

**Macrophage immunofluorescence**. Before immunofluorescent assay, antigen retrieval, fluorescence cancellation, and serum blocking were performed. To stain macrophages after 1 week and 2 weeks of implantation, sections were incubated with a rabbit anti-mouse F4/80 monoclonal antibody (dilution 1:400; catalog no. 30325 from Cell Signaling Technology) overnight at 4 °C. After being washed three times with phosphate-buffered saline (PBS) the sections were incubated with Alexa Fluor® 488-conjugated goat anti-rabbit IgG (H + L) antibody (dilution 1:400; catalog no. GB25303 from Servicebio) for 50 min in dark. The sections were washed three times with PBS and incubated with DAPI (1 µg/mL in PBS) for 10 min. The macrophages show green fluorescence and the nucleus shows blue fluorescence.

**Vascular density analysis**. The blood vessel density after 4 weeks and 3 months of implantation were evaluated using immunofluorescent labeling. Sections were incubated with mouse anti-mouse αSMA monoclonal antibody (dilution 1:500; catalog no. GB13044 from Servicebio) overnight at 4 °C. After being washed three times with PBS the sections were incubated with Alexa Fluor® 488-conjugated goat anti-mouse IgG (H + L) antibody (dilution 1:400; catalog no. GB25301 from Servicebio) for 50 min in dark. The sections were washed three times with PBS and incubated with DAPI (1 µg/mL in PBS) for 10 min. The vascular endothelial cells show green fluorescence and the nucleus show blue fluorescence. The amount of blood vessels around each hydrogel (within 100 µm from the hydrogel-tissue interface) was counted ($n = 6$, mean values ± sd).

**Immunohistochemical staining of inflammatory factors**. The rabbit anti-mouse CCR7 monoclonal antibody is from Novus Biologicals (dilution 1:200; catalog no. NBP2-67324). The rabbit anti-mouse TNF-α polyclonal antibody is from Servicebio (dilution 1:400; catalog no. GB11188). The goat anti-mouse IL-12 polyclonal antibody is from Novus Biologicals (dilution 1:100; catalog no. NB600-1443). The goat anti-mouse IL-10 polyclonal antibody is from Novus Biologicals (dilution 1:100; catalog no. AF519). Before immunoassay, antigen retrieval, endogenous peroxidase cancellation, and bovine serum albumin blocking were performed sequentially. Sections of samples 2 weeks post-implantation were incubated with a primary antibody overnight at 4 °C. The sections were washed three times with PBS and incubated with HRP-labeled goat anti-rabbit antibody or HRP-labeled rabbit anti-goat antibody (1:200; catalog no. GB23303 or GB23204 from Servicebio) at room temperature for 50 min in dark. Sections were washed three times, dried slightly, and then incubated with fresh prepared Diaminobenzidine (DAB) chromogenic reagent Kit (DAKO, catalog no. K5007). The sections were counterstained in the nucleus with Hematoxylin staining solution (Servicebio, catalog no. G1004) for 3 min and wash with water. Cells stained by inflammatory markers show brown color, while all nucleus stained with hematoxylin show blue color. The number of positive expressions around each hydrogel (within 100 µm from the hydrogel-tissue interface) was counted ($n = 6$, mean values ± s.d.).

**Protein extraction**. Proteins in the tissues surrounding the Mock group without implantation, PEG hydrogels, and PSer hydrogels were extracted using a full protein extraction kit (Solarbio, catalog no. BC3710) 2 weeks post-implantation. In brief, 1 mL of cold RIPA lysate (containing phosphatase inhibitor, protease inhibitor, and phenylmethylsulfonyl fluoride) was added to the shredded tissue, and the mixture was grinded at 4 °C. After the tissue lysate was centrifuged at 12,000 g for 30 min at 4 °C, the supernatant was pipetted into a new tube. The extracted protein is stored in aliquots at −80 °C, and the protein concentration is measured by the BCA kit (Beyotime, catalog no. P0012) before use.

**Cytokine profiling analysis**. Proteome profiler antibody array (R&D System, catalog no. ARY006) were used for cytokine profiling. 2 mL of blocking buffer were pipetted into each well of the 4-well multi-dish containing each antibody array membranes and incubated for 1 h on a shaker. The blocking buffer was aspirated and 1.5 mL sample solutions, contain 300 µg proteins and 15 µL detection antibody cocktail (positive control), was pipetted to each well and incubated overnight at 4 °C. Each membrane was carefully placed into individual plastic containers and washed thrice by 1× wash buffer. The membrane was put into the 4-well multi-

dish, containing 2 mL diluted Streptavidin-HRP (1:2000), and incubated for 30 min on a shaker. After washed thrice by 1× wash buffer, the membranes were incubated with 1 mL of Chemi Reagent Mix for 1 min and placed in an autoradiography film cassette (ImageQuant LAS 4000, GE Healthcare) for chemiluminiscence. The gray value was measured by ImageQuant LAS 4000 Control software.

**RNA-seq and data analysis**. Before the experiment, all appliances were treated by diethyl pyrocarbonate (DEPC) for RNA-DNase free. After implantation of hydrogels for 2 weeks, tissue samples from implantation were cut into small pieces and immersed into RNA-later solution, using a Mock group without implantation as the control. RNA samples were qualified and quantified using a NanoDrop and Agilent 2100 bioanalyzer (Thermo Fisher Scientific, MA, USA). The RNA sequencing libraries were constructed and sequenced on the BGISEQ-500 platforms.

**qPCR assay**. Reverse transcription was performed to create cDNA using Prime-Script™ RT reagent Kit (TaKaRa, catalog no. RR037A). cDNA (1 µL; 1:20 dilution) in a total volume of 20 µL (including TB Green, TaKaRa, catalog no. RR420A, and PCR primers) was amplified by qPCR with the following primers. Primers (Supplementary Table 5) were designed and evaluated using the NCBI Primer-BLAST tool to ensure mouse specificity. Samples were incubated at 95 °C for 10 min followed by 44 cycles at 95 °C for 15 s and 60 °C for 1 min in a CFX96 Real-Time System (with Bio-Rad CFX Manager software). The fold increase or decrease was determined relative to a Mock control after normalizing using the $2^{-\Delta\Delta CT}$ method ($n = 4$, mean values ± sd).

**Statistics and reproducibility**. Statistical analysis was performed with Origin software. Significance between the two groups was determined by a two-tailed t-test. One-way analysis of variance (ANOVA) with Tukey post-test for more than two variables was carried out. All results were expressed as mean ± standard error. All micrograph assays were carried out at least three independent times with similar results.

**Reporting summary**. Further information on research design is available in the Nature Research Reporting Summary linked to this article.

## Data availability

Data that support the findings detailed in this study are available in the Supplementary Information and this article. RNA-seq data have been deposited in the NCBI Gene Expression Omnibus database under accession code PRJNA699958. Any other source data perceived as pertinent are available, on reasonable request, from the corresponding author. The Source data underlying Fig. 4b, c, d, f, g, h, 5b, d, 6a, b, c, d, e, g, Supplementary Figures. 3b, 4, 5c, and 7 are provided as a Source data file. Source data are provided with this paper.

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

## Acknowledgements

We thank Prof. Aiguo Hu, Dr. Peng Zhao, and Yusen Wu at East China University of Science and Technology for their help on GPC characterization. We thank Prof. Linyong Zhu and Yiqing Zhang at East China University of Science and Technology for their help on compressive strength measurement. The authors also thank the Research Center of Analysis and Test of East China University of Science and Technology for the help on the characterization. Funding: This work was supported by the National Natural Science Foundation of China for Innovative Research Groups (No. 51621002), the National Natural Science Foundation of China (No. 21774031, 31800801), the Natural Science Foundation of Shanghai (18ZR1410300), the National Key Research and Development Program of China (2016YFC1100401), Program of Shanghai Academic/Technology Research Leader (20XD1421400), Research Program of State Key Laboratory of Bioreactor Engineering, the Fundamental Research Funds for the Central Universities.

## Author contributions

R.L. directed the whole project. D.Z. and R.L. conceived the idea, proposed the strategy, designed the experiments, evaluated the data, and wrote the manuscript together. D.Z. performed the majority of the experiments. Q.C. participated in implantation experiments, immunohistochemical analysis, and genetic analysis. Y.B. and C.S. participated in the hydrogels preparation and implantation experiments. H.Z. and J.W. participated in the implantation experiments. M.C. drew schematic diagrams and participated in data analysis. W.Z. participated in data analysis. J.Z. operated the circular dichroism test. Z.Q. operated the SEM. J.L. participated in implantation experiments and relevant data analysis. S.C. participated in result discussion and troubleshooting. All authors proofread the paper.

## Competing interests

R.L. and D.Z. are co-inventors on a patent application covering reported materials and application to resist the FBR. All remaining authors declare no competing interests.
