## [Peer Review File · Nature Communications]

REVIEWER COMMENTS

Reviewer #1 (Remarks to the Author):

The idea of utilizing a natural material-similar polymer is good. However, low immunogenicity does not equal low FBR. Sericin protein is known to induce a macrophage response. From the Masson's trichrome staining of the encapsulations of both PEG and Pser group, there is no obvious difference. In Fig. 4a, the red layer between the hydrogel and fibrosis tissue is not a dense capsule as the writer pointed out. In some cases, this red cell layer between implant and fibrosis tissue may be dominated by neutrophils and macrophages indicating acute inflammation due in part to infection. Dense capsule should display a blue color using Masson's trichrome staining, see references below [1-3]. In Fig. 4d, both Pser 3300 and Pser 8600 show a certain thickness of fibrous encapsulation, but the authors quantify them as zero. Therefore, the accuracy of capsule quantification thickness in Fig.4 is in question. As

the interpretations are not accurate to prove the capability of this material, it is recommended not to publish this work at this time.

1. Farah, S., Doloff, J.C., Müller, P. et al. Long-term implant fibrosis prevention in rodents and non-human primates using crystallized drug formulations. *Nat. Mater.* 2019, 18, 892–904.

2. Rujitanaroj PO, Jao B, Yang J, et al. Controlling fibrous capsule formation through long-term down-regulation of collagen type I (COL1A1) expression by nanofiber-mediated siRNA gene silencing. *Acta Biomaterialia.* 2013 Jan;9(1):4513-4524.

3. M.N. Avula, A.N. Rao, L.D. McGill, D.W. Grainger, F. Solzbacher, Foreign body response to subcutaneous biomaterial implants in a mast cell-deficient Kitw-Shmurine model, *Acta Biomaterialia.*, 2014, 10(5) 1856-1863.

Reviewer #2 (Remarks to the Author):

February 6, 2021

Referee report

Bio-inspired poly-DL-serine materials resist the foreign-body response

This is a relevant, interesting and detailed study on a new class on implants that are resistant to the foreign body response (FRB). However, there are a few points that should be addressed before acceptance.

General points:

The immunogenicity and antigenicity of PEG and PEG-modified objects is alluded to in the introduction. It should be no surprise that a material that activates complement would give a more vigorous reaction than one that does not activate the complement system. This point should be addressed in the conclusions. The study would have been improved if a polymer like a polyacrylamide gel was used as a control, as it has similar water content, but is not particularly a complement activator.

Another stimulator of local reaction to implants is the mechanics of the implant. A stiffer implant will generate more reaction. Compare the mechanical properties of the PEG gels to the Pser gels.

Line 38 The word "urgent" should be removed. Such materials that do not generate a FBR might help improve outcomes with some types of implants, but millions of implants of "ordinary" biomaterials are used each year with reasonable success. It is hardly urgent to have these materials.

Line 61 This sentence does not make sense in this paper: "but it is highly desired to design anti-FBR materials using components that exit in the body to ensure high biocompatibility and low

immunogenicity.” There is no data here on PSer biodegradation. Furthermore, the concept is to have the implant stay in place for long periods without the FBR. In some cases, biodegradability is good, in other cases it is undesirable.

Line 134 “PEG hydrogels were surrounded by a dense collagen capsule, around 27 μm for PEG2000 and around 22 μm for PEG5000.” Capsules of 27 μm and 22 μm would not be considered dense. For most implants of “biocompatible” biomaterials, capsules are 80-150 microns in thickness, but still satisfy regulatory agency requirements for biocompatibility. The PEG reaction is actually quite mild (but the PSer is milder). Remove the word “dense.”

Line 195 I would not refer to TIMP1 as a fibrosis protein. It inhibits MMPs, but that does not make it a fibrosis protein.

Line 360+ I could find no description on how the capsule thickness was assessed? What identification criteria, sampling protocols and statistics were used? This is controversial so it should be elaborated upon.

Line 373+ Similar to the comment above, how was blood vessel counting and quantification performed?

Point-by-point response to the reviewers' comments

REVIEWER COMMENTS

Reviewer #1 (Remarks to the Author):

The idea of utilizing a natural material-similar polymer is good. However, low immunogenicity does not equal low FBR. Sericin protein is known to induce a macrophage response. From the Masson's trichrome staining of the encapsulations of both PEG and Pser group, there is no obvious difference. In Fig. 4a, the red layer between the hydrogel and fibrosis tissue is not a dense capsule as the writer pointed out. In some cases, this red cell layer between implant and fibrosis tissue may be dominated by neutrophils and macrophages indicating acute inflammation due in part to infection. Dense capsule should display a blue color using Masson's trichrome staining, see references below [1-3]. In Fig. 4d, both Pser 3300 and Pser 8600 show a certain thickness of fibrous encapsulation, but the authors quantify them as zero. Therefore, the accuracy of capsule quantification thickness in Fig.4 is in question. As the interpretations are not accurate to prove the capability of this material, it is recommended not to publish this work at this time.

1. Farah, S., Doloff, J.C., Müller, P. et al. Long-term implant fibrosis prevention in rodents and non-human primates using crystallized drug formulations. *Nat. Mater.* 2019, 18, 892–904.

2. Rujitanaroj PO, Jao B, Yang J, et al. Controlling fibrous capsule formation through long-term down-regulation of collagen type I (COL1A1) expression by nanofiber-mediated siRNA gene silencing. *Acta Biomaterialia.* 2013 Jan;9(1):4513-4524.

3. M.N. Avula, A.N. Rao, L.D. McGill, D.W. Grainger, F. Solzbacher, Foreign body response to subcutaneous biomaterial implants in a mast cell-deficient Kitw-Shmurine model, *Acta Biomaterialia.*, 2014, 10(5) 1856-1863.

Response: We are very grateful to the reviewer's valuable comments. We have made substantial changes according to these comments.

We agree with the reviewer's comment that "low immunogenicity does not equal low FBR." Indeed, the FBR is an immunological process, where inflammatory-immunologic reactions take place in the earliest stages of a response, promoting subsequent pro-fibrotic processes (*Annu. Rev. Immunol.* **2013**, 31, 107). Efforts to attenuate the host response to implants have led to the development of immune-isolating materials (*Trends Biotechnol.* **2016**, 34, 470). For example, Prof. Shaoyi Jiang and coworkers reported that low FBR zwitterionic poly(carboxybetaine) is a low immunogenic polymer (*Angew. Chem. Int. Ed.* **2018**, 57, 13873). Given the strong indications that mucins hold immune-modulating capacities, Prof. Thomas Crouzier and coworkers developed a mucin hydrogel that can evade fibrotic FBR (*Adv. Funct. Mater.* **2019**, 29, 1902581). Therefore, implanted materials with low immunogenicity are likely to cause low inflammation and finally result in low FBR, which inspired us to design low FBR materials from materials with low immunogenicity.

Some literatures reported that sericin protein could induce a macrophage response

for vascularization and wound healing (*Sci. Rep.* **2019**, *9*, 3448; *ACS Biomater. Sci. Eng.* **2020**, *6*, 3502). Nevertheless, study of the macrophage response of silk protein concludes that sericin usually does not manifest inflammatory activity when present in the soluble form, and macrophage activation study of silk protein shows that sericin when attached to fibers induces inflammatory responses (*Biomaterials* **2003**, *24*, 3079; *Prog. Polym. Sci.* **2008**, *33*, 998). Multiple literatures also reported that sericin protein has very low or no immunogenicity (*J. Control. Release* **2006**, *115*, 307; *J. Chem. Technol. Biotechnol.* **2006**, *81*, 136; *J. Biosci. Bioeng.* **2009**, *107*, 556; *Biotechnol. Adv.* **2015**, *33*, 1855). It is clarified in the research paper from Prof. David Kaplan and co-workers that “Soluble sericin proteins extracted from native silk fibers did not induce significant macrophage activation. While sericin did not activate macrophages by itself, it demonstrated a synergistic effect with bacterial lipopolysaccharide” (*Biomaterials* **2003**, *24*, 3079). The research from Prof. Yuqing Zhang and co-workers demonstrated that “The immunogenicity of silk sericin and the antigenicity of sericin-insulin derivatives were not observed in both rabbits and mice” (*J. Control. Release* **2006**, *115*, 307). Further study from this group reported that “The silk sericin peptides have no immunogenicity, and the antigenicity of the L-asparaginase is obviously decreased when coupled covalently with the silk sericin peptides” (*J. Chem. Technol. Biotechnol.* **2006**, *81*, 136). Prof. Pornanong Aramwit and co-workers reported that “The maximum levels of TNF- α and IL-1 β released from monocytes and macrophage cells after sericin induction would not cause an inflammatory response or prevent cellular proliferation” (*J. Biosci. Bioeng.* **2009**, *107*, 556), and “Silk sericin was found to suppress the production of pro-inflammatory cytokines” (*Nat. Prod. Commun.* **2013**, *8*, 501). In a recent report from Prof. David Kaplan and coworkers, they gave a summary that “While there has been some confusion over time as to the true source of the silk’s allergic response, recent careful studies have come to the consensus that the allergic response is elicited by the native combined fibroin/sericin structure, but that either fibroin or sericin alone does not elicit an allergic reaction” (*Biomaterials* **2015**, *71*, 145). Many other researchers also recognized that sericin on its own is low or no immunogenicity (*Acta Biomater.* **2009**, *5*, 3007; *Sci. Rep.* **2014**, *4*, 7064; *Acta Biomater.* **2016**, *41*, 210; *ACS Appl. Mater. Interfaces* **2017**, *9*, 3432; Bio-response to Silk Sericin. Chapter: 11. In: Kundu S, ed. *Silk Biomaterials for Tissue Engineering and Regenerative Medicine*. Woodhead Publishing, **2014**, *74*, 299). Considering all these precedent literatures and information, we hypothesized that sericin protein is likely to have low immunogenicity. Given hydrophobic and cationic moieties seem to promote more inflammatory response further than hydrophilic moieties (*J. Mater. Sci. Mater. Med.* **2015**, *26*, 121; *J. Biomed. Mater. Res. A* **2017**, *105*, 927), we designed PSer that composed of the hydrophilic serine without those hydrophobic and cationic moieties. Our result in this study showed that sericin-inspired poly-DL-serine (PSer) does display low immunogenicity and low FBR as implanted hydrogels.

We thank the reviewer for the comment and provided literatures regarding the Masson’s trichrome staining of the encapsulations of PEG and PSer group in our study, which inspired us to read this part in precedent literatures more carefully, including three references provided by the reviewer. We also added these three papers as

references (Ref. 24-26) into our revised manuscript. In fact, an evaluation on collagen density can better reflect the degree of the FBR as used in precedent studies from Prof. Shaoyi Jiang group (*Nat. Biotechnol.* **2013**, *31*, 553-556), from Prof. Minglin Ma group (*Biomaterials* **2020**, *230*, 119640) and from Prof. Robert Langer/Daniel Anderson group (*Nat. Biotechnol.* **2016**, *34*, 345). These recent literatures and studies suggest that collagen density is a more appropriate evaluation on FBR than does collagen thickness. Therefore, in our revised manuscript, we removed the collagen thickness measurement and measured the collagen density at the interface (within 50 μm , at 10- μm steps) from the hydrogel-tissue interface, after implantation for 4 weeks (Fig. 4c) and 3 months (Fig. 4g), respectively. The collagen density at the interface of the two PSer hydrogels and tissues (62-76%) was significantly lower than that of both PEG hydrogels (>90%) after implantation for 4 weeks (Fig. 4c in our revised manuscript, as shown below). After implantation for 3 months, both PEG hydrogels were still encapsulated by a dense capsule layer with >90% collagen density, while PSer hydrogels were surrounded by diffused collagens at a density of 48-69% (Fig. 4g in our revised manuscript, as shown below).

Fig. 4. Collagen and blood vessel density in tissues surrounding the PEG and PSer hydrogels 4 weeks and 3 months post subcutaneous implantation.

We thank the reviewer for the comment on the red layer between the hydrogel and fibrosis tissue in Fig. 4a of our original manuscript. We agree with the reviewer on this

and changed description of the red layer between the hydrogel and fibrosis tissue from “the collagen capsule thickness” to “the inflammatory cell thickness” in Fig. 4b and 4f in our revised manuscript, by following the explanation in precedent literatures (*J. Biomed. Mater. Res. A* **2010**, 93, 941; *Biomaterials* **2015**, 41, 26; *Biomaterials* **2015**, 41, 79). In our revision, we also made changes accordingly from “...were surrounded by a dense collagen capsule, ... both PSer hydrogels were surrounded by tissues with only a thin capsule layer ...” to “...were surrounded by a dense layer of inflammatory cells, ... both PSer hydrogels were surrounded by tissues with only a thin layer of inflammatory cells ...”, and made changes from “...were still encapsulated by a dense capsule layer..., while no obvious dense capsule layer was found...” to “...were still encapsulated by a dense layer of inflammatory cells..., while no obvious layer of inflammatory cells was found...”.

Reviewer #2 (Remarks to the Author):

February 6, 2021

Referee report

Bio-inspired poly-DL-serine materials resist the foreign-body response

This is a relevant, interesting and detailed study on a new class on implants that are resistant to the foreign body response (FRB). However, there are a few points that should be addressed before acceptance.

Response: We thank the reviewer for the favorable comments and inspiring questions below.

General points:

The immunogenicity and antigenicity of PEG and PEG-modified objects is alluded to in the introduction. It should be no surprise that a material that activates complement would give a more vigorous reaction than one that does not activate the complement system. This point should be addressed in the conclusions. The study would have been improved if a polymer like a polyacrylamide gel was used as a control, as it has similar water content, but is not particularly a complement activator.

Response: We thank the reviewer for this suggestion. We synthesized two polyacrylamide hydrogels (PAM1 and PAM2) using 20 wt% precursor, so the hydrogels have water content similar to PEG and PSer hydrogels, which were synthesized using 20 wt% precursor. PAM1 was synthesized by 19 wt% acrylamide and 1 wt% bisacrylamide; PAM2 was synthesized by 18 wt% acrylamide and 2 wt% bisacrylamide (Supplementary Table 2.). After subcutaneous implantation in mice, both PAM hydrogels (PAM1 and PAM2) cause obvious inflammatory response after 1 week and collagen capsulation after 4 weeks, as showed in the histological staining (H&E for

1 week and M&T for 4 weeks, Supplementary Fig. 6 in our revised manuscript). Therefore, despite the polyacrylamide hydrogel is not a complement activator, it caused obvious FBR after implantation. Our observation is consistent to other researchers' report on the foreign-body granuloma of the glabella due to polyacrylamide filler (*Case Rep. Dermatol.* **2013**, 5, 181-185). In short, by following the suggestion from the reviewer and add extra control of PAM hydrogels, we further strengthened our conclusion and improved our manuscript. We have added related results and discussions into our revised manuscript.

Supplementary Figure 6. (a) Explantation picture and H&E staining of polyacrylamide (PAM) hydrogels after subcutaneously implanted in mice for 1 week. (b) M&T staining of PAM hydrogels after 4 weeks implantation.

Another stimulator of local reaction to implants is the mechanics of the implant. A stiffer implant will generate more reaction. Compare the mechanical properties of the PEG gels to the PSer gels.

Response: We agree with the reviewer's comment. We measured the mechanical properties of PEG and PSer hydrogels by compressive test. The result showed that PEG hydrogels have stronger compression modulus (1.07 MPa for PEG2000 and 0.582 MPa for PEG5000) than PSer hydrogels (0.121 MPa for PSer3300 and 0.046 MPa for PSer8600) (Supplementary Fig. 3 in our revised manuscript as shown below), though we prepared all these hydrogels using 20 wt% PSerDA and PEGDA solutions.

Beyond the reviewer's request, we also prepared even softer hydrogels, PEG5000-H hydrogel, using only 4 wt% of PEG5000DA as precursor. The PEG5000-H hydrogel has a high water content (~96.0%) and a low compressive modulus (0.017 MPa), which is much weaker than both PSer hydrogels (Supplementary Fig. 3-4 in our revised manuscript as shown below). After 1 week implantation of PEG5000-H hydrogels, the hydrogels were wrapped with a layer of inflammatory cells, which is weaker than PEG2000 and PEG5000 in causing the body's inflammatory response. Although the mechanical properties of PEG5000-H hydrogels are only 1/7.2 of PSer3300 and 1/2.8 of PSer8600, the inflammatory response caused by PEG5000-H hydrogels are still stronger than that of both PSer hydrogels (Supplementary Fig. 5a in our revised manuscript and image comparison shown below). After 4 weeks implantation, PEG5000-H hydrogels were encapsulated by a dense layer of inflammatory cells with fibrosis, resulted in obvious FBR (Supplementary Fig. 5b in our revised manuscript and

result comparison shown below). Quantification on the collagen density at the interface of the PEG5000-H hydrogels and tissues gave a ratio of 87-91% that is similar to the collagen density around PEG2000 and PEG5000 hydrogels and is much higher than the collagen density around Pser3300 and Pser8600 hydrogels (Supplementary Fig. 5c in our revised manuscript and result comparison shown below). Therefore, above studies on the mechanics of the implant, using the stiffness, support our conclusion and the merit of our Pser hydrogels as promising anti-FBR materials. The effect of stiffness on anti-FBR effect between different types of materials in our observation is also supported by what has been reported on PHEMA and PCBMA hydrogels in anti-FBR study (*Nat. Biotechnol.* **2013**, *31*, 553-556).

Supplementary Figure 3. (a) Compressive curves of hydrogels. (b) Compressive modulus of hydrogels.

Supplementary Figure 4. Water content of hydrogels.

Supplementary Figure 5. (a) Explantation picture and H&E staining of PEG5000-H hydrogels after subcutaneously implanted in mice for 1 week. (b,c) M&T staining (b) and quantified collagen density (c) of PEG5000-H hydrogel-tissue interface after 4 weeks of implantation.

Image comparison (H&E staining) of PSer3300, PSer8600 and PEG5000-H hydrogels after implantation for 1 week.

Result comparison of PSer3300, PSer8600 and PEG5000-H hydrogels after implantation for 4 weeks. (a) M&T staining. (b) Collagen density. # $p < 0.05$, using one-way ANOVA with Tukey post-test.

Line 38 The word “urgent” should be removed. Such materials that do not generate a FBR might help improve outcomes with some types of implants, but millions of implants of “ordinary” biomaterials are used each year with reasonable success. It is hardly urgent to have these materials.

Response: We agree with the reviewer’s comment and have removed the word “urgent” from our text in the revised manuscript.

Line 61 This sentence does not make sense in this paper: “but it is highly desired to design anti-FBR materials using components that exit in the body to ensure high biocompatibility and low immunogenicity.” There is no data here on PSer biodegradation. Furthermore, the concept is to have the implant stay in place for long periods without the FBR. In some cases, biodegradability is good, in other cases it is undesirable.

Response: We agree with the reviewer that “*In some cases, biodegradability is good, in other cases it is undesirable*”. In fact, the information we try to deliver in this sentence is our hypothesis to design anti-FBR materials using components that exit in the body to ensure low immunogenicity. We didn’t mean to favor only the biodegradable materials for our purpose. Nevertheless, our original expression may not clear enough and leads to confusion. So, we removed this sentence in our revised manuscript to avoid misunderstanding from readers.

Line 134 “PEG hydrogels were surrounded by a dense collagen capsule, around 27 μm for PEG2000 and around 22 μm for PEG5000.” Capsules of 27 μm and 22 μm would not be considered dense. For most implants of “biocompatible” biomaterials, capsules are 80-150 microns in thickness, but still satisfy regulatory agency requirements for biocompatibility. The PEG reaction is actually quite mild (but the PSer is milder). Remove the word “dense.”

Response: We thank the reviewer for this comment. Comparing to the evaluation of collagen thickness, an evaluation on collagen density can better reflect the degree of the FBR as used in precedent studies from Prof. Shaoyi Jiang group (*Nat. Biotechnol.* **2013**, *31*, 553-556), from Prof. Minglin Ma group (*Biomaterials* **2020**, *230*, 119640) and from Prof. Robert Langer/Daniel Anderson group (*Nat. Biotechnol.* **2016**, *34*, 345). These recent literatures and studies suggest that collagen density is a more appropriate evaluation on FBR than does collagen thickness. Therefore, in our revised manuscript, we removed the collagen thickness measurement and measured the collagen density at the interface (within 50 μm , at 10- μm steps) from the hydrogel-tissue interface, after implantation for 4 weeks (Fig. 4c) and 3 months (Fig. 4g), respectively. The collagen density at the interface of the two PSer hydrogels and tissues (62-76%) was significantly lower than that of both PEG hydrogels (>90%) after implantation for 4 weeks (Fig. 4c in our revised manuscript, as shown below). After implantation for 3

months, both PEG hydrogels were still encapsulated by a dense capsule layer with >90% collagen density, while Pser hydrogels were surrounded by diffused collagens at a density of 48-69% (Fig. 4g in our revised manuscript, as shown below).

Fig. 4. Collagen and blood vessel density in tissues surrounding the PEG and Pser hydrogels 4 weeks and 3 months post subcutaneous implantation.

Line 195 I would not refer to TIMP1 as a fibrosis protein. It inhibits MMPs, but that does not make it a fibrosis protein.

Response: We thank the reviewer for point out of this. We have modified our description on this from "...lower expression of the fibrosis protein TIMP-1," to "...lower expression of the TIMP-1 that inhibits matrix metalloproteinase and promotes fibrosis," in our revised manuscript.

Line 360+ I could find no description on how the capsule thickness was assessed? What

identification criteria, sampling protocols and statistics were used? This is controversial so it should be elaborated upon.

Response: We thank the reviewer for this question. We used inflammatory cell thickness and collagen density instead of capsule thickness to get a better evaluation of the FBR. We added detailed information in the Methods part: “The inflammatory cell thickness is measured according to the thickness of markedly red to purple layer at the hydrogel-tissue interface in the M&T images (n = 6, mean values \pm s.d.). The collagen density is measured by the percentage of blue-pixel coverage in the M&T images of tissues within 50 μm (at 10- μm steps) from the hydrogel-tissue interface (n = 6, mean values \pm s.d.).” The statistical analysis was described in the Methods part that “Significance between two groups was determined by two-tailed t-test. One-way analysis of variance (ANOVA) with Tukey post-test for more than two variables was carried out. All results were expressed as mean \pm standard error.” The statistical analysis was also described independently in the legends of Figures.

Line 373+ Similar to the comment above, how was blood vessel counting and quantification performed?

Response: We thank the reviewer for this question. We added detailed information in the Methods part: “The vascular endothelial cells show green fluorescence and the nucleus show blue fluorescence. The amount of blood vessels around each hydrogel (within 100 μm from the hydrogel-tissue interface) was counted (n = 6, mean values \pm s.d.).”

We greatly thank all the reviewers' valuable comments. We hope that the revised manuscript will prove to be acceptable for publication in *Nature Communications*.

Sincerely,

Runhui Liu

Professor of Chemistry and Biomaterials

REVIEWER COMMENTS

Reviewer #1 (Remarks to the Author):

Basically the author addressed the issues as requested before. However, for one data interpretation regarding the inflammatory cell thickness, to my knowledge, people generally use H&E stained sections for this measurement instead of Masson's trichrome stained sections. If the author can provide relevant information regarding the methods and data of this, I think this manuscript is good to go.

Reviewer #2 (Remarks to the Author):

I am satisfied with the conscientious response to my comments. The manuscript is improved with the revisions.

Point-by-point response to the reviewers' comments

REVIEWER COMMENTS

Reviewer #1 (Remarks to the Author):

Basically the author addressed the issues as requested before. However, for one data interpretation regarding the inflammatory cell thickness, to my knowledge, people generally use H&E stained sections for this measurement instead of Masson's trichrome stained sections. If the author can provide relevant information regarding the methods and data of this, I think this manuscript is good to go.

Response: We thank the reviewer for pointing out this and previous suggestions to help us improve our manuscript. We used Masson's trichrome (M&T) stained sections to quantify the inflammatory cell layer, according to the method in precedent literature (*Acta Biomater.* **2019**, *100*, 105). We choose M&T staining, though H&E staining is widely used, because we used one staining to quantify both the thickness of inflammatory cell layer and the collagen density. Since the inflammatory cell layer can be clearly distinguished in both H&E and M&T stains, the evaluation of the inflammatory cell layer by the two staining methods is consistent, though difference in color.

For your convenience to compare these two staining methods, we also did H&E staining on two representative sections, as shown below. The PEG2000 hydrogel implantation for 4 weeks showed a thick layer of inflammatory cells. The Pser8600 hydrogel implantation for 3 months showed negligible inflammatory cell layer. These demonstrations indicate that both M&T and H&E staining give the same evaluation and conclusion on the thickness of inflammatory cell layer in our study.

Reviewer #2 (Remarks to the Author):

I am satisfied with the conscientious response to my comments. The manuscript is improved with the revisions.

Response: We thank the reviewer for the positive response and suggestions to help us improve our manuscript.

We greatly appreciate all the reviewers' valuable comments. We hope that the revised manuscript will prove to be acceptable for publication in *Nature Communications*.

Sincerely,

Runhui Liu

Professor of Chemistry and Biomaterials